# Sensory Symptoms without Structural Pathology in Patients with Gluten Sensitivity

**DOI:** 10.3390/nu16081209

**Published:** 2024-04-19

**Authors:** Marios Hadjivassiliou, Nick Trott, Nigel Hoggard, David S. Sanders

**Affiliations:** 1Academic Department of Neurosciences, Sheffield Teaching Hospitals NHS Trust, Glossop Road, Sheffield S10 2JF, UK; 2Department of Dietetics, Sheffield Teaching Hospitals NHS Trust, Sheffield S10 2JF, UK; nick.trott@nhs.net; 3Department of Infection, Immunity and Cardiovascular Disease, University of Sheffield, Sheffield S10 2RX, UK; 4Academic Department of Gastroenterology, Sheffield Teaching Hospitals NHS Trust, Sheffield S10 2JF, UK

**Keywords:** gluten sensitivity, coeliac disease, neurological manifestations, sensory symptoms, gluten-free diet

## Abstract

We report on a group of patients with gluten sensitivity with and without coeliac disease presenting with unexplained sensory symptoms in the absence of structural pathology. Methods: The patients were selected from the gluten neurology clinic based at the Royal Hallamshire Hospital, Sheffield, UK, on the basis of sensory symptoms but normal neuroaxis imaging and peripheral nerve evaluation. Results: A total of 30 patients were identified with a mean age at presentation of 47 years. The prevalence of enteropathy was 78%. The sensory disturbance was characterised by tingling at 50%, numbness at 27%, pain at 20%, burning at 13% and “buzzing” feeling at 7%. The distribution of the sensory symptoms included hands and feet in 27% of the patients, torso in 27%, legs only in 23%, face in 17% and arms only in 10%. For five patients, the sensory disturbance was migratory and affected different parts of the body at any given time. After the introduction of a gluten-free diet, 77% of patients noted significant improvement in their sensory symptoms. In one-third of the patients, there was a complete resolution of the sensory symptoms. Conclusion: Unexplained sensory symptoms can be seen in patients with gluten sensitivity and respond to strict adherence to a gluten-free diet.

## 1. Introduction

Gluten sensitivity (GS) encompasses a spectrum of disorders triggered by the ingestion of gluten in genetically susceptible individuals [1]. The best characterised entity within this spectrum is coeliac disease (CD), also known as gluten sensitive enteropathy [2]. Patients with GS can present with gastrointestinal symptoms (bloating, abdominal pain, diarrhoea, constipation) but extraintestinal manifestations can also be very common and can often be the presenting feature. Patients with GS do not always have CD. They may have serological positivity for gluten sensitivity-related antibodies (e.g., antigliadin antibodies—AGAs) in the absence of enteropathy [1].

The commonest neurological manifestation of GS is cerebellar ataxia (gluten ataxia) with the second most common being peripheral neuropathy (gluten neuropathy) [1]. The types of neuropathies encountered in the context of GS include sensorimotor axonal length-dependent neuropathy (the commonest), sensory ganglionopathy and small fibre neuropathy [1,3,4]. Small fibre neuropathy can be present in isolation or in combination with the other two types. The third common manifestation of gluten sensitivity is gluten encephalopathy [1]. Gluten encephalopathy refers to a combination of frequent headaches associated with slow cognition (some patients refer to this as “foggy brain”) that often interferes with everyday activities. Such patients often have evidence of increased white matter lesions on MR imaging. Characteristically, these patients respond very well to a strict gluten-free diet (GFD).

A range of antibodies can be used in the context of diagnosing GS and CD. Endomysium (EMA) and tissue transglutaminase antibodies (TGG) are specific to the presence of enteropathy (CD) [2]. AGAs may be better markers of the whole spectrum of GS whilst TG6 antibodies have been shown to be a more specific marker for the neurological manifestations of GS with and without CD [5,6,7].

The gluten neurology clinic based at the Academic Department of Neurosciences, Sheffield Teaching Hospitals NHS Trust has been caring for patients with gluten sensitivity-related neurological dysfunction over the last 27 years. Patients have either already been diagnosed with coeliac disease and are referred to this clinic due to the development of neurological symptoms, or have existing neurological dysfunction (e.g., ataxia, neuropathy) and thus undergo extensive serological screening for gluten sensitivity including AGAs (IgG and IgA), TGG, EMA and TG6 antibodies (IgA and IgG). If one or more of these antibodies are positive and other causes of the neurological problem have been excluded, a diagnosis of gluten-related neurological dysfunction is made. A total of 1888 such patients have been seen and assessed in this specialist clinic over the last 27 years.

During this time, we have made the observation that some patients complain of sensory symptoms yet, despite extensive investigations for both central and peripheral pathology, have no evidence of a neuropathy or any other central nervous system abnormalities to explain the sensory symptoms. We describe here a cohort of patients with gluten sensitivity and unexplained sensory symptoms and report the effect of a gluten-free diet as well as discussing possible pathophysiology.

## 2. Materials and Methods

The gluten sensitivity/neurology clinic, based at the Royal Hallamshire Hospital, Sheffield Teaching Hospitals NHS Trust, was set up in 1996. It cares for over 1888 patients with GS with or without CD who present with neurological symptoms or who are known to have CD and subsequently develop neurological manifestations. The diagnosis of gluten sensitivity-related neurological dysfunction was based on either a pre-existing diagnosis of CD and the development of neurological dysfunction or otherwise unexplained neurological dysfunction and positive serology for GS. Serological testing included AGAs (IgG and IgA), TGG, EMA and TG6 antibodies (IgA and IgG). Whilst caring for this large cohort of patients, it became apparent that a number of patients complained of sensory symptoms, yet extensive investigations including neurophysiological assessments, including small fibre neuropathy tests (nerve conduction studies, thermal thresholds and SUDOSCAN testing) as well as brain and spinal imaging failed to identify any pathology to explain these sensory symptoms. We describe the typical presentation, type and distribution of sensory disturbance as well as the effect of the gluten-free diet on this cohort of patients.

## 3. Results

### 3.1. Clinical Characteristics and Investigations

A total of 30 patients were identified. Twenty-one (70%) were female and nine male. The mean age at the time of presentation was 47 years (range 23–75). In 20/30 (67%) of the patients, the original presentation was with neurological symptoms. Serological testing for gluten sensitivity-related antibodies at presentation showed the following: 94% positive for TG6 antibodies (as these antibodies were introduced in clinical practice in 2019, only 17 of the 30 patients were tested at baseline), 63% positive for AGAs, 43% positive for TG2 antibodies and 30% positive for endomysium antibodies. In 43%, more than one of the above antibodies were positive. The prevalence of enteropathy (villous atrophy, crypt hyperplasia and increased intraepithelial lymphocytes) amongst 18 patients who underwent gastroscopy and duodenal biopsy was 14/18 (78%). The reasons why 12 patients did not have a biopsy included personal choice (refused to have a gastroscopy), the pandemic (availability of gastroscopy and biopsy during the pandemic was extremely limited) and negative serology for CD (EMA and TG2 antibodies), which decreased the likelihood of an enteropathy. There was a good correlation between positivity for TG2 and/or EMA and the presence of enteropathy. There were just two out of fourteen patients with enteropathy who were negative for EMA and TG2. Both were positive for TG6 antibodies. 

All patients included in the series had neurophysiological assessments assessing large fibres (nerve conduction studies and electromyography), thermal thresholds and SUDOSCAN testing to assess small fibre involvement. All patients had normal neurophysiological assessments. In addition, these patients underwent brain and spinal imaging using an MRI scan. Apart from some evidence of cerebellar involvement in 10 patients (those with gait ataxia), the imaging was normal. Additional neurological features were seen in half of these patients. The commonest additional features were cerebellar ataxia (10) or headaches (5). Bowel symptoms were present in 13 patients, 10 of which had enteropathy on biopsy. 

### 3.2. Sensory Symptoms

The sensory disturbance was characterised by tingling in 50% of the patients, numbness in 27%, pain in 20%, burning in 13% and a “buzzing” feeling in 7%. A third of the patients had a combination of more than one of these sensory symptoms. The distribution of the sensory symptoms was as follows: hands and feet in 27% of the patients, torso in 27%, legs only in 23%, face in 17% and arms only in 10%. In 60% of the patients, the sensory disturbance affected more than one of the above body areas. In five patients, the sensory disturbance was migratory and affected different parts of the body at any given time. The severity of the sensory symptoms was such that it necessitated referral to neurology by the patients’ general practitioner, gastroenterologist or neurologist.

### 3.3. Effect of Gluten-Free Diet

All 30 patients received dietetic input and advice for GFD. Within a year of the introduction of the gluten-free diet, 23/30 (77%) of the patients noted significant improvement in their sensory symptoms. In 11 out of the 23 patients that improved, there was complete resolution of the sensory symptoms. The remaining six patients said that the sensory symptoms remained the same. One patient was lost to follow-up. Several patients disclosed that the sensory symptoms recured if they inadvertently ingested gluten (see illustrative cases below). In one patient, the sensory symptoms took several years to completely resolve.

Serological testing at the last clinic visit showed that 14 patients were still positive for AGA (8/14) and TG6 antibodies (10/14), with five being positive for both. None of the patients were positive for EMA or TG2 antibodies. 

Two cases selected from this cohort of 30 patients are described in more detail below in an attempt to illustrate the typical clinical presentation and the spectrum of the presenting features as well as the effect of GFD in alleviating the symptoms.

The clinical characteristics, serological profile and the effect of GFD are summarised in Table 1.

### 3.4. Case 1

A 58-year-old lady was referred by her GP to the gluten sensitivity neurology clinic. She had been previously assessed (6 years prior) by another neurologist due to an altered sensation initially affecting her face and then hands and feet. Initially, the sensory disturbance (tingling and numbness) was transient but then became more permanent, which is why she was referred for further evaluation. Previous investigations that included brain imaging and neurophysiology were normal. The patient made the observation that the symptoms were worse or sometimes triggered by eating bread or pasta. This was partly the reason why she was referred to the gluten sensitivity neurology clinic. She put herself on a gluten-free diet and noticed substantial improvement in her sensory symptoms.

Upon examination during the first encounter, there was no demonstrable sensory deficit but the patient had difficulty tandem walking. All blood tests were normal including vitamin B12, folic acid, copper and vitamin E. Serological testing for gluten sensitivity was negative for EMA and TG2, with a borderline antigliadin antibody level (upper limit of normal). The tests were conducted whilst the patient was on a strict gluten-free diet. Repeat neurophysiological assessments including small fibre assessment (thermal thresholds and SUDOSCAN) were normal. Spinal imaging was normal but brain imaging showed reduced spectroscopic measurements from the cerebellar vermis (low N acetyl-aspartate to creatine-NAA/Cr) in keeping with her mild gait ataxia. There was no cerebellar atrophy. The patient was reluctant to re-expose herself to gluten in order to be retested for gluten sensitivity and CD. She was therefore seen by our expert dietitian who confirmed that the patient had good knowledge of the gluten-free diet and that she was very vigilant in avoiding gluten. She was given some further advice in avoiding cross-contamination.

She was followed-up at regular intervals and her serology was repeated each time she attended. The patient commented that, if she inadvertently had some gluten exposure (usually when eating out), she noticed the sensory symptoms being very prominent and more extensive. There was always a background residual tingling sensation in her hands and feet. Repeat brain imaging with MR spectroscopy of the cerebellum a year after further dietary advice showed improvement in the NAA/Cr from the vermis. The gluten serology remained negative.

During a review in one of her regular appointments, 4 years after the initial visit, she complained that her sensory symptoms had suddenly become much worse. She also noticed a numbness affecting her tongue. This deterioration coincided with her eating a cake in a café that was incorrectly labelled as being gluten-free. The sensory symptoms became much worse within 24 h of the exposure. Serological testing 3 weeks after exposure showed very high levels of antigliadin antibodies (IgA and IgG), whilst EMA and TG2 antibodies remained negative. Repeat brain imaging showed no changes to the previous scan and nothing to account for the numbness in her tongue. Her symptoms again subsided during the next 2 months. She remains stable although some minor sensory symptoms (tingling in her toes and fingers) are present all the time.

### 3.5. Case 2

A 56-year-old lady was acutely admitted at a district general hospital with a 6-week history of tingling affecting her hands and feet. She complained of a headache and word-finding difficulties. She had no previous medical history and no bowel complaints. Upon examination, there was no evidence of any focal pathology; in particular, there was no sensory loss, reflexes were preserved and plantars were flexor. Blood investigations (blood count, urea and electrolytes, liver and thyroid function tests, vitamin B12, folic acid and vitamin E, autoimmune profile) showed no abnormality. Her case was discussed with the regional neurology on-call team who advised spinal and brain imaging. This was performed and was normal. She underwent a lumbar puncture. Cerebrospinal fluid (CSF) examination was normal. In particular, the CSF protein was normal. She was transferred to the regional neurology unit to undergo further evaluation. The patient had a degree of gait ataxia but the examination was otherwise normal.

Nerve conduction studies and electromyography were normal. Gangioside antibodies including GQ1b were negative. The thermal threshold study was normal.

MR spectroscopy of the cerebellum showed normal structures but reduced NAA/Cr over the cerebellar vermis and hemispheres. She underwent physiotherapy and was started on medication (pregabalin) for neuropathic pain. As part of the screening tests for what was presumed to be a neuropathy, she was tested for coeliac disease. She was positive for EMA (no other serology was performed). A duodenal biopsy confirmed the presence of CD and she was given advice for a GFD.

The widespread sensory symptoms continued and a further duodenal biopsy was performed a year later. This showed improvement but still some increase in the intraepithelial lymphocytes.

She was reviewed on a 6-monthly basis and the antibody profile was repeated. EMA, TG2 and AGA antibodies were negative. A further neurophysiological assessment, including thermal threshold testing one year later, was again normal. She underwent SUDOSCAN testing which was also normal.

Three years after the original presentation, the patient reported some improvement in her sensory symptoms and in her balance. She managed to come off pregabalin and continued to be assessed by the physiotherapists. She reported that, on two occasions when she inadvertently consumed gluten, she felt sick immediately afterwards. Repeat brain imaging including MR spectroscopy of the cerebellum showed significant improvement in the NAA/Cr measurement of the vermis and cerebellar hemisphere.

Five years after the original presentation, she reported minimal tingling in her hands and feet, normal balance and no gastrointestinal symptoms. At the last clinic review, 8 years since the original presentation, she reported that the sensory symptoms had now completely subsided and that this was the best she had ever been. Serological testing (AGA, TGG, EMA and TG6 antibodies) was negative.

## 4. Discussion

We describe here a cohort of patients with gluten sensitivity, with or without enteropathy presenting with sensory symptoms and no evidence of structural pathology of the central and peripheral nervous systems to explain these. Sensory symptoms in the context of GS and CD can be seen in gluten neuropathy. Indeed, large epidemiological studies have shown that coeliac disease is associated with 2.5-fold increased risk of neuropathy [8]. The cohort described here, however, had no evidence of a peripheral neuropathy despite extensive neurophysiological assessments evaluating both large and small fibres. In the absence of any evidence of peripheral nerve dysfunction, we postulate that such symptoms are more likely to be centrally mediated. The absence of any structural pathology on brain and spinal imaging does not necessarily argue against central pathology. Furthermore, the improvement in the sensory symptoms following a strict GFD in 77% of these patients suggests that these symptoms are linked to GS. 

Previous work by our group demonstrated the involvement of the thalamus amongst newly diagnosed patients with CD who were positive for TG6 antibodies [7]. Using normalising brain volumetry (NBV), we were able to show significantly reduced grey matter in the thalamus and the cerebellum on what otherwise appeared to be a structurally normal brain. The thalamus is a major relay station for sensory processing and its involvement in these patients could potentially suggest central pathology as an explanation for these sensory symptoms. More evidence that such symptoms are likely to be centrally mediated comes from the fact that, in most cases described here, the sensory symptoms are transient and, in some cases, migratory. Indeed, five patients with migratory sensory symptoms could be labelled as having Wartenberg’s migrant sensory neuritis [9]. This is a condition of unknown aetiology, characterised by sudden patchy numbness affecting different parts of the body with a tendency of altered sensory deficit distribution over time. Neurophysiological assessments are usually normal. It is likely that the original description by Wartenberg represented a heterogeneous group of disorders but the symptoms described in his original publication were not dissimilar to what we have observed in 5 out of our 30 patients. Whilst in most patients the presentation is gradual, in one of our patients (case 2), the presentation was much more acute resulting in acute admission to hospital for urgent investigations. In this particular case, the response to the GFD was much more protracted and it took over 5 years for the patient to fully recover.

The pathophysiology of coeliac disease and the neurological dysfunction in gluten sensitivity remain partially unresolved, although some facts have been established [10]. Firstly, there is evidence of an immune-mediated neurological damage, as shown by the inflammatory changes seen on brain tissue and peripheral nerve from patients with gluten ataxia and gluten neuropathy [1]. Secondly, there is evidence to suggest that transglutaminase 6 is implicated in the pathophysiology of gluten ataxia and other neurological manifestations [5]. This is analogous to transglutaminase 2 (tissue transglutaminase), the autoantigen in coeliac disease and transglutaminase 3 (epidermal transglutaminase), the autoantigen in dermatitis herpetiformis, the skin manifestation of gluten sensitivity [11,12]. TG2, TG3 and TG6 have 65% homology, all are capable of deamidating gliadin peptides (a crucial step in the production of immunogenic peptides) and antibodies against these transglutaminases are eliminated with strict adherence to a gluten-free diet [13]. It has been shown that patients with neurological manifestations of gluten sensitivity have serological evidence of TG6 antibodies. Indeed, in the group of patients described here, 94% were positive for TG6 antibodies, although baseline TG6 antibody testing was only available in 17 patients. Amongst patients with a classic presentation of coeliac disease, 40% also had TG6 antibodies. We have demonstrated that this 40% represents patients that are more susceptible to neurological dysfunction, such as having significantly reduced grey matter in the thalamus and the cerebellum [7]. 

Identifying this group of patients with unexplained sensory symptoms in our clinic has considerably helped in their management. These patients have sometimes been labelled as having functional neurological disorder (FND), as all investigations come back normal or negative [14]. It is important to perform these investigations (brain and spinal imaging and neurophysiology) in symptomatic patients with established gluten sensitivity, given that CNS and peripheral nerve involvement are very common in the context of gluten sensitivity (gluten ataxia, gluten neuropathy, gluten encephalopathy). Equally important is to perform appropriate serological testing not just for coeliac disease (TGG and EMA), but also for gluten sensitivity (AGA and TG6 antibodies), given that the presence of enteropathy is not a prerequisite for developing neurological dysfunction. As shown in these series of patients, the prevalence of enteropathy was 78% and the response to the gluten-free diet was equally favourable irrespective of the presence of enteropathy. Our approach, therefore, is to firstly complete the investigations and then explain to the patient that such unexplained sensory symptoms can be seen in the context of gluten sensitivity and respond to a strict adherence to a gluten-free diet. This information often comes as a relief for these patients who are understandably concerned about the very real sensory symptoms they are experiencing.

## 5. Conclusions

In summary, we present here a cohort of patients with gluten sensitivity and unexplained sensory symptoms responding to a strict gluten-free diet. Clinicians should be aware of such presentation in order to be able to correctly diagnose gluten sensitivity with or without CD and treat such patients. It is also imperative for the correct serological testing to be performed for the accurate diagnosis of the whole spectrum of gluten-related disorders.

## Figures and Tables

**Table 1 nutrients-16-01209-t001:** Clinical characteristics and effect of gluten-free diet (AGA antigliadin antibodies, TG6 transglutaminase 6, TGG tissue transglutaminase, EMA endomysium antibody). F, female; M, male.

	Total Number of Patients = 30						
Mean age at presentation	47 years (range 23–75)						
F:M ratio	21:9						
Number presenting with neurological symptoms	67%						
Serology at baseline	94% TG6 antibody positive	63% AGA positive	43% TGG antibody positive	30% EMA positive	43% had more than one positive antibody		
Enteropathy	14/18 (78%) of those that underwent biopsy						
Sensory symptoms	tingling 50%	numbness 27%	pain 20%	burning 13%	buzzing 7%		
Distribution of sensory symptoms	hands and feet 27%	torso 27%	legs only 23%	face 17%	arms only 10%	migratory 17%	more than one part of the body 60%
Effect of gluten-free diet	23/30 (77%) improved	11/23 complete resolution	6/30 stable	1 lost to follow-up			

## Data Availability

The original contributions presented in the study are included in the article, further inquiries can be directed to the corresponding author.

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
