# Peer review of "Sensory Symptoms without Structural Pathology in Patients with Gluten Sensitivity"

_nutrients, 2024, doi:10.3390/nu16081209_

Round 1

Reviewer 1 Report

Comments and Suggestions for Authors

The paper describes neurological symptoms in patients with gluten sensitivity. The research results seem interesting, but in my opinion they do not fit the profile of the journal at all.

Moreover, the work contains a number of shortcomings. The study group is poorly characterized. The results obtained are only given in general terms, described statistically, and there is no correlation between individual parameters.

The paper is inconsistent because it describes the results of a cohort of patients and two case reports. The literature is sparse, contains old references and is poorly selected (36% are works by the main author).

Author Response

We thank the reviewer for the helpful comments. We have tried to address these as follows:

The paper describes neurological symptoms in patients with gluten sensitivity. The research results seem interesting, but in my opinion they do not fit the profile of the journal at all.

We are pleased that the reviewer finds our results interesting. As indicated in the paper these are novel observations (never been described before) in patients with coeliac disease and/or gluten sensitivity.The paper was submitted at the invitation of the Editor of a special theme issue of Nutrients “Food Intolerance and food allegy: novel aspects in a changing world”. Both the Editor of this special issue and the Editor of Nutrients indicated that this report does indeed fit the profile of the journal, which is why it was send for reviewing.

Moreover, the work contains a number of shortcomings. The study group is poorly characterized. The results obtained are only given in general terms, described statistically, and there is no correlation between individual parameters.

We are not entirely clear of the specific issues raised by the reviewer here. This is descriptive work, rather than a study, but all the clinical information about the cohort of 30 patients has been provided. We have added a table with all the clinical characteristics of this cohort of patients for further clarity. We have also clarified further the selection process in the materials and methods section.

The paper is inconsistent because it describes the results of a cohort of patients and two case reports. The literature is sparse, contains old references and is poorly selected (36% are works by the main author).

The 2 case reports were deliberately selected from the cohort of 30 patients to try and illustrate the spectrum of the clinical presentations in these patients. We selected one patient with coeliac disease who presented acutely with sensory symptoms and another with gluten sensitivity (no CD) whose presentation was much more subacute. The 2 cases, we felt, were representative of this cohort and give a flavour of the typical presentation. We have justified the selection of the 2 patients in the text (page 7).

We have included references that include peripheral neuropathy (the only previously described gluten-related condition resulting in sensory symptoms) as these, we feel, are the only ones of any relevance to this paper. We have added some additional references that will inevitably reduce the percentage of self referencing. The Editor had already asked us to remove some of the references referring to our own work to make it acceptable for the Journal. We have already done so.

Reviewer 2 Report

Comments and Suggestions for Authors

The paper is very interesting since is focused on a clinical manifestation of gluten-sensitivity not extensively studied. Some issues should be addressed by authors:

Introduction: A more specific explanation of features and serological characteristics of CS should be added in order to better understand the study design/results (TG6, TG6 antibodies..). Some more references should be added

Materials and methods: The selection of patients should be better expressed

Results: I suggest to add some graphs/tables representing patients'characteristics and the main results. In addition, the choice of cases (n. 1 and 2) should be clarified

Discussion: A more extensive parison with literature data should be reported

Author Response

Reviewer 2

We thank the reviewer for the helpful comments. We have tried to address these as follows:

The paper is very interesting since is focused on a clinical manifestation of gluten-sensitivity not extensively studied. Some issues should be addressed by authors

We thank the reviewer for their positive comment.

Introduction: A more specific explanation of features and serological characteristics of CS should be added in order to better understand the study design/results (TG6, TG6 antibodies..). Some more references should be added

We have added 2 paragraphs (page 3 and page 4) to explain more about the antibodies and the total cohort of patients from which this group has been selected. We have moved forward the references about TG6 antibodies (reference 5 and 7) and added another one on the subject (reference 6).

Materials and methods: The selection of patients should be better expressed

We have elaborated on this within this section.

Results: I suggest to add some graphs/tables representing patients'characteristics and the main results. In addition, the choice of cases (n. 1 and 2) should be clarified

We have added a table summarising the results (table 1, page 7). The choice of the 2 patients was based on selecting representative examples, one with a relatively acute presentation who was then diagnosed with CD (case 2) and the other with a more chronic presentation who was gluten sensitive and did not have an enteropathy (case 1). This patient also exhibited the typical recurrence of sensory symptoms after exposure to gluten. We have justified the selection of the 2 patients in the text (page 7).

Discussion: A more extensive parison with literature data should be reported

We have expanded the discussion, as suggested, and added some relevant references. We have also clarified the TG6 antibody positivity at baseline. This test was not available for all as it was introduced in 2019 for clinical use.

Round 2

Reviewer 1 Report

Comments and Suggestions for Authors

Thank you very much for your substantive responses to my review. The authors' answers are convincing. As I have already assessed, the work is original and innovative. The authors clarified my doubts about the studied groups and presented the results of investigations in a more clear way. Due to the fact that the work presents original and new research results on a group of patients with gluten sensitivity (with or without celiac disease) with unexplained sensory symptoms in the absence of structural changes, I propose to accept the work.